# Oral Supplementation with Hydrolyzed Fish Cartilage Improves the Morphological and Structural Characteristics of the Skin: A Double-Blind, Placebo-Controlled Clinical Study

**DOI:** 10.3390/molecules26164880

**Published:** 2021-08-12

**Authors:** Patrícia Maria Berardo Gonçalves Maia Campos, Rodolfo Scarpino Barboza Franco, Letícia Kakuda, Gabriel Fernandes Cadioli, Gabriela Maria D’Angelo Costa, Elodie Bouvret

**Affiliations:** 1School of Pharmaceutical Sciences of Ribeirão Preto, University of São Paulo, Av do Café s/n, Ribeirão Preto 14040-903, SP, Brazil; rodolfosbfranco@gmail.com (R.S.B.F.); leticia.kakuda@usp.br (L.K.); gabriel.cadioli@usp.br (G.F.C.); gmdcosta@usp.br (G.M.D.C.); 2Abyss Ingredients, 860 Route de Caudan, 56850 Caudan, France; elodie@abyss-ingredients.com

**Keywords:** skin aging, hydrolyzed fish cartilage, nutricosmetics, imaging analysis, clinical efficacy

## Abstract

Collagen and its peptides are natural ingredients used in food supplements and nutricosmetics with the claim of providing benefits for skin health and beauty. In this context, the aim of the present study was to evaluate the clinical efficacy of oral supplementation with hydrolyzed fish cartilage for the improvement of chronological and photoaging-induced skin changes. A total of 46 healthy females aged 45 to 59 years were enrolled and divided into two groups: G1—placebo and G2—oral treatment with hydrolyzed fish cartilage. Measurements of skin wrinkles, dermis echogenicity and thickness, and morphological and structural characteristics of the skin were performed in the nasolabial region of the face before and after a 90-day period of treatment using high-resolution imaging, ultrasound, and reflectance confocal microscopy image analyses. A significant reduction in wrinkles and an increase of dermis echogenicity were observed after a 90-day period of treatment with hydrolyzed fish cartilage compared to the placebo and baseline values. In addition, reflectance confocal microscopy (RCM) image analysis showed improved collagen morphology and reduced elastosis after treatment with hydrolyzed fish cartilage. The present study showed the clinical benefits for the skin obtained with oral supplementation with a low dose of collagen peptides from hydrolyzed fish cartilage.

## 1. Introduction

Exposure to exposome accelerates the skin aging process by reducing cellular metabolism, promoting the degradation of macromolecules such as collagen, and decreasing natural physiological responses, especially attenuating the antioxidant response. These changes reflect on the skin and contribute to premature skin aging [1,2].

Among the components of this exposure, solar radiation significantly contributes to accelerated skin aging by promoting photoaging [2]. UV radiation increases the formation of free radicals in the skin, causing DNA damage and expression of matrix metalloproteinases responsible for the degradation of extracellular matrix proteins such as collagen, elastin, and hyaluronic acid [3,4,5]. These changes are clinically perceived by an increase in skin dryness, alterations in skin elasticity and microrelief, and the appearance of wrinkles [6,7,8].

Thus, topical and oral treatments have been used for the improvement of skin conditions. Topical treatment includes the use of cosmetic products based on active ingredients for skin hydration and protection as well as for the improvement of cutaneous changes induced by photoaging [8].

In this context, consumption of nutraceuticals as functional foods and oral supplements has been used to improve skin conditions [9]. Collagen and its peptides are natural ingredients used in various food supplements and nutraceuticals with the claim of providing benefits for skin health and beauty among other applications [7,8,9,10,11].

Clinical studies have shown that dietary supplementation can modulate skin functions and improve skin health and beauty [8,9,10,11,12]. Oral supplementation with collagen peptides has shown more pronounced effects on the skin compared to topical products [8].

Some types of hydrolyzed collagen from different sources have been proposed for skin care such as bovine, porcine, chicken, and fish collagen. Type I collagen is the most common type of product used for health care and skin beauty, while type II collagen is preferred and more used for joint health. In addition, it is common to combine collagen peptides with glycoaminoglycans and micronutrients such as vitamin C and minerals [13].

Several studies have described the mechanism of absorption and distribution of collagen peptides in the body. However, the dosage used for hydrolyzed collagen from chicken or porcine sources can be considered high, from 2.5 g to 10 g per day, which can compromise adherence to long-term treatment with collagen peptides [6,7,8,9,10,11,12]. In addition, collagen has shown problems of solubility in water due to its high molecular weight, more than 300 kDa, which can influence administration depending on the proposed dosage [14]. Thus, oral supplementation with lower concentrations of hydrolyzed collagen and evaluation of its clinical efficacy for the skin are very important topics.

Noninvasive biophysical and imaging techniques have been widely used in clinical studies that assess the clinical changes of aged skin by instrumental measurements. These techniques analyze a variety of skin-related factors, and the correlation of the results obtained provides a broad view of the mechanical, morphological, and structural characteristics of the skin in a noninvasive way. These techniques contribute to the assessment of the efficacy of oral and topical treatments of the skin, especially regarding the dermis [2,7,8,9,10,15].

Reflectance confocal microscopy (RCM) is an advanced imaging technique that allows the study of the morphology and structure of the epidermis at a close histological level [2,5,16]. Dermal changes related to aging can also be assessed by this technique [17,18]. The use of RCM is an innovative proposal for the evaluation of an oral hydrolyzed fish cartilage supplement. This technique, combined with a high frequency ultrasound skin imaging system and high definition imaging resolution, allows a complete assessment of collagen morphology and echogenicity before and after the use of oral supplementation.

In this context, the aim of the present study was to evaluate, using skin imaging techniques, the clinical efficacy of oral supplementation with hydrolyzed fish cartilage for the improvement of cutaneous photoaging changes. Finally, the present study involved a new approach to collagen supplementation by assessing the clinical benefits for the skin of low-dosage oral supplementation with hydrolyzed fish cartilage.

## 2. Results

### 2.1. Skin Microrelief

An improvement in skin microrelief was observed after supplementation with hydrolyzed fish cartilage (Figure 1b). However, this result was not significant (Figure 1a), probably due to interindividual differences.

### 2.2. Dermis Echogenicity and Thickness

The dermis echogenicity ratio (number of low echogenical pixels divided by number of total echogenical pixels—LEP/TEP) decreased after the 90-day period of supplementation with hydrolyzed fish cartilage. Thus, a significant increase in dermis echogenicity was observed due to a reduction of the hypoechogenic pixel/total number of pixel ratio compared to baseline values. In addition, a significant and pronounced improvement in dermis echogenicity was observed after 90 days of supplementation with hydrolyzed fish cartilage compared to the placebo (Figure 2b) (*p* < 0.05). The increase in dermal echogenicity suggests an improvement in dermis density [8].

Figure 2a illustrates the improvement of dermis echogenicity in a participant who received hydrolyzed fish cartilage supplementation for 90 days compared to the placebo.

After the 90-day period of supplementation with hydrolyzed fish cartilage, a significant increase in dermis thickness was observed compared to baseline values and to the placebo (Figure 2c) (*p* < 0.05), suggesting an improvement in dermis hydration.

### 2.3. Skin Wrinkles and Pores Examined by High Definition Imaging Resolution

Figure 3 presents the high-resolution images of the skin before and after 90 days of supplementation. It is possible to observe an improvement of skin wrinkles on the frontal, nasolabial, and periorbital regions of the face in G2. In addition, Figure 4 shows the wrinkle score (%) for the regions of the face in the study groups before (baseline) and after supplementation.

In summary, analysis by the scoring method of Group 2 showed a reduction in skin wrinkles in the frontal (14% ± 0.68) region of the face and a significant reduction in the wrinkles in the nasolabial (31% ± 0.81) when compared the T0 and T90, and periorbital (26% ± 0.66) regions of the face (Figure 3 and Figure 4) when compared the T0 and T90 of the G2 group and with the placebo at T90.

Figure 5b shows high-resolution images with fine pores outlined in green and large pores in red on the malar region of the face before and after 90 days of the study. Figure 5a shows the pore score for the malar region of the face of the participants before and after 90 days of supplementation, showing a reduction in the number of fine and large pores for the G2 group after 90 days. The results of pore scores in mean ± standard deviation were as follows: G1 T0 = 4.5 ± 0.60; G1 T90 = 4.7 ± 0.46; G2 T0 = 4.8 ± 0.40; and G2 T0 = 4.6 ± 0.50.

Analysis by the scoring method showed a significant reduction in skin wrinkles in the periorbital and nasolabial regions of the face after supplementation with hydrolyzed fish cartilage (Figure 3 and Figure 4) (*p* < 0.05).

### 2.4. Morphological and Structural Skin Characteristics Determined by Reflectance Confocal Microscopy (RCM)

The results of epidermis thickness obtained for the placebo group and hydrolyzed fish cartilage group are listed in Table 1. No significant difference in the RCM parameters evaluated was observed between groups or times of study. However, the stratum corneum thickness was significantly increased after 90 days of treatment with the hydrolyzed fish cartilage.

The results of the RCM descriptors of photoaged skin of the G1 and G2 groups are listed in Table 2. G2—Hydrolyzed fish cartilage showed a statistically significant difference in irregular honeycomb pattern, low interkeratinocyte reflectance, elastosis, coarse and huddled collagen structures after a 90-day period of treatment compared to initial time (T0). In addition, there was a statistically significant difference between the T0 and T90 times in the irregular honeycomb pattern and low interkeratinocyte reflectance for the placebo group—G1.

RCM imaging analysis revealed signs of photoaging in the epidermis and dermis of both study groups since changes were observed in the morphology of dermal architecture, dermal epidermal junction, and epidermis honeycomb pattern before the treatments. After 90 days of supplementation, RCM imaging analysis showed an improvement in photoaging changes for the group supplemented with hydrolyzed fish cartilage.

The parameters of irregular honeycomb pattern were reduced by 44% and 60% in G1 and G2, respectively, after a 90-day period of supplementation. The parameter of low interkeratinocyte reflectance was reduced by 48% and 100% after a 90-day period of supplementation in the placebo group and in the hydrolyzed fish cartilage group, respectively. These findings may have been associated with skin hydration [16].

The dermis morphology score obtained by the RCM image analysis is shown in Figure 6 through the relative frequency of the scores. The results in mean ± standard deviation were as follows: G1 T0 = 4.0 ± 0.9; G1 T90 = 4.0 ± 0.9; G2 T0 = 4.0 ± 0.7; and G2 T90 = 3.0 ± 0.0. There was an increase of score 3 related to the presence of coarse collagen for G2 after the 90-day period, whereas no change in score was observed in G1 after 90 days of the study.

There was an improvement in honeycomb pattern and an increase in interkerationocyte brightness in all study groups after the 90-day period (Figure 7). However, this result was more pronounced in the group treated with hydrolyzed fish cartilage. The morphology of the dermal epidermal junction was unchanged after both supplementations (Figure 7). The morphological pattern of the dermis improved after the 90-day period in the group supplemented with hydrolyzed fish cartilage—G2. There was a 100% reduction of huddled collagen and a 20% increase in coarse collagen after the 90-day period in G2, in addition to a 19% reduction of elastosis (Figure 7).

In summary, Figure 8 illustrates representative images showing a reduction in wrinkles, dermis echogenicity, and collagen morphology improvement after a 90-day period of treatment with hydrolyzed fish cartilage.

### 2.5. Perceived Efficacy

After the 90 day period of study, the participants were asked about their perception of the efficacy of product application to the skin. Most of the participants who received the hydrolyzed fish cartilage supplement reported improvement of skin tonus, firmness, hydration, and wrinkles, effects that were less perceived by the placebo group (Figure 9).

## 3. Discussion

The aging process changes the characteristics of the skin, inducing wrinkles and reducing dermal density, among other clinical changes. Thus, oral collagen supplementation has been used to reduce the effects of skin aging and to improve skin conditions. In this context, we report here the clinical effects on mature skin of supplementation with low dosage collagen peptides obtained from hydrolyzed fish cartilage (500 mg/day) using noninvasive skin imaging techniques. This supplement showed an improvement in skin microrelief, dermis echogenicity, and collagen structure as well as a significant reduction in wrinkles, effects that were perceived by the study participants.

Considering that collagen digestion occurs by the action of peptidases in stomach and the peptides are absorbed in the intestinal epithelium in the form of various collagen peptides [19,20], the ingestion of hydrolyzed collagen can bring benefits for skin health. However, the bioavailability of oral supplementation with collagen or hydrolysates depends on the type and source of this collagen [21]. In addition, bioavailability depends on the molecular weight of the collagen, and marine sources generally have a low molecular weight, which could improve absorption [13]. Thus, the benefits of oral supplementation with hydrolyzed fish cartilage could be associated with its bioavailability when compared to other collagen sources [13,22].

In a review study, Vollmer et al., (2018) [13] reported that oral administration of collagen peptides generally improves the characteristics of aged skin. In addition, Schwartz and Park (2012) [23], Genovese et al., (2017) [11], and Zmitek et al., (2020) [6] reported an improvement in the skin parameters related to skin microrelief and reduced facial aging signs associated with an increase in skin hydration after oral supplementation with hydrolyzed collagen.

In our study, imaging analysis showed an improvement in skin microrelief after supplementation with hydrolyzed fish cartilage due to an increase in the Sew parameter, which represents the number and width of wrinkles. However, this improvement was not statistically significant, probably due to interindividual variation.

On the other hand, high-resolution imaging analysis showed a significant reduction in wrinkles in the periorbital and nasolabial regions of the skin after 90 days of supplementation with hydrolyzed fish cartilage. In addition, a reduction in wrinkles was also observed in the frontal region. Thus, the results obtained showed that hydrolyzed fish cartilage was effective in wrinkle reduction, an effect probably due to improved dermal density. Some studies carried out on animal models and in in vitro tests have shown that the administration of collagen peptides can increase dermal density and contribute to its homeostasis [13,24,25].

High-resolution analysis combined with a score method was also used here to evaluate skin pores. The results showed that hydrolyzed fish cartilage supplementation was effective in reducing skin pores, in agreement with Lin et al., (2020) [26], who showed that oral supplementation with collagen can be effective in reducing pores and wrinkles.

We also observed that supplementation with hydrolyzed fish cartilage promoted a significant decrease in dermis echogenicity in the nasolabial region of the face. This result revealed an increase in dermis density, greater formation of collagen fibrils, and repair of current damage, helping to reduce the chronological aging and photoaging process and improving the skin density [8,27]. An increase in dermis thickness was also observed and was correlated with hydration of deeper skin layers and with the maintenance of skin homeostasis, thus helping to reduce the damage caused by chronological aging and other factors [6,7,8].

According to the results obtained by RCM imaging analysis, supplementation with hydrolyzed fish cartilage improved the morphological and structural characteristics of the dermis when compared to the baseline and placebo, since it increased coarse collagen and reduced elastosis. Score analysis supported this finding since there was an increase in coarse collagen (score 3) and a decrease in huddled collagen and elastosis (score 5) after the 90-day period of oral use of the hydrolyzed fish cartilage supplement, indicating an improved collagen pattern.

Solar elastosis appears in a severe photoaging process, and its reduction is a good indication that photoaging has improved. Additionally, huddled collagen occurs when the individual fibers are no longer visible and is composed of amorphous hyporeflective material. Its presence characterizes an advanced aging process, so that its reduction or disappearance is a good sign. Coarse collagen is common in the age group used in the present study and is characterized by thick fibers coarsely arranged to form a network [17,18].

Czajka et al. (2018) [7] studied the clinical efficacy of three months of the oral supplementation of collagen peptides combined with vitamins and other bioactive agents on aged skin using histological analyses of the forearm skin of four subjects in order to evaluate the morphological characteristics of the aged skin before and after supplementation. The results showed an improvement in the structure and stratification of the epidermal layers, in the organization of collagen fibers in the dermis, an increase in the thickness of the papillary dermis and in the number of fibrocytes, and a reduction of elastosis [7]. In our study, RCM permitted the evaluation of the morphological structural characteristics of the skin in a non-invasive way as an alternative to invasive histological analyses that cause discomfort to the patient. In addition, the results corroborated the findings of Czajka et al. (2018) [7], who showed improvement in collagen morphology and a reduction in elastosis after daily oral supplementation with a nutraceutical containing hydrolyzed fish collagen, vitamins, antioxidants, and other active ingredients.

RCM analysis showed an improvement of the RCM descriptors related to the dermis in the group that used the oral hydrolyzed fish cartilage supplement, in agreement with the results obtained in the analysis of dermis echogenicity, which was improved after the use of the hydrolyzed fish cartilage supplement for 90 days. In addition, although there have been some studies showing an improvement in dermis echogenicity after supplementation with collagen peptides [8,12,26], the application of RCM to the evaluation of skin morphological and structural characteristics after oral supplementation has not been previously investigated. Thus, RCM collagen analysis can be used as a supporting method for dermis evaluation in combination with other imaging techniques such as high frequency ultrasound analysis.

The results obtained in this study corroborate other studies in the scientific literature that report the benefits of oral supplementation of hydrolyzed collagen in improving the characteristics of skin aging [28]. However, the limitations of the study was the difficulty in controlling and monitoring the participants’ diet, especially in terms of their protein intake, which could have led to uneven levels of protein absorption resulting from different diet habits.

In conclusion, supplementation with the hydrolyzed fish cartilage acted in depth on the skin due to an increase in dermis echogenicity, suggesting an improvement in dermis density. The RCM images showed that the hydrolyzed fish cartilage supplement improved the morphological and structural characteristics of the dermis by improving the collagen morphology. In addition, the study participants also perceived the results obtained by the high-resolution image analysis, since they were able to observe an improvement in the skin parameters evaluated. In the clinical evaluation, the participants also perceived a reduction in wrinkles and an improvement in skin texture as well as a moisturizing effect and an improvement in skin firmness and appearance.

Finally, the present study showed important clinical benefits for the skin with a lower dosage (500 mg per day) of oral supplementation with hydrolyzed fish cartilage.

## 4. Materials and Methods

### 4.1. Study Design

A randomized, double-blind, and placebo-controlled study was conducted after approval by the Ethics Committee for Clinical Research of the School of Pharmaceutical Sciences of Ribeirão Preto/SP (CEP/FCFRP no. 439. CAAE no. 65109317.2.0000.5403). The study was carried out in accordance with ICH Guidelines on Good Clinical Practice and the Declaration of Helsinki [29,30]. All participants gave their written informed consent to participate in the study.

### 4.2. Study Participants

Forty-six healthy females were recruited based on the inclusion/exclusion criteria. Inclusion criteria were healthy Caucasian women, aged 45–59 years (mean age: 52 ± standard deviation: 5), Fitzpatrick Skin Phototypes II and III, and Corporal Mass Indicator from 18 to 25 kg/m^2^ and the presence of wrinkles on the face. Exclusion criteria were pregnancy, smokers, diabetics, addicted to alcohol or other narcotic drugs, history of adverse reaction to cosmetics, systemic diseases, skin disease, and use of any cosmetics or supplementation during the study.

The participants were instructed not to use cosmetic products two weeks before and during the study period, except for a sunscreen (SPF 30+), which was applied once a day in the morning All participants agreed to avoid being exposed in the sun during the period of the study.

A total of 46 participants were enrolled in the study and 43 concluded it (Figure 10). Three study participants dropped out because one moved to another city and two traveled during the study.

### 4.3. Study Products and Intervention

Participants were randomized using a computerized random number into two groups. After randomization, the echogenicity of the dermis was measured using the Dermascan^®^ equipment. The data obtained were submitted to statistical analysis to confirm that the study groups did not differ significantly from each other. Then, the participants received the placebo or the hydrolyzed fish cartilage according to the groups below.

Group 1 (G1)—Placebo: 23 healthy participants received and ingested one capsule containing 500 mg of maltodextrin, once a day, every night during a period of 90 days.

Group 2 (G2)—Hydrolyzed fish cartilage: 23 healthy women received and ingested one capsule containing 500 mg of hydrolyzed fish cartilage, once a day, every night during a period of 90 days.

The hydrolyzed fish cartilage (Cartidyss^®^ NG, Abyss Ingredients, Caudan, France) is a water-soluble powder obtained by standardized enzymatic hydrolysis of fish cartilage without preservatives or processing aids. From the natural composition of raw material, the hydrolyzed fish cartilage contains more than 65% of collagen peptides, at least 95% of which have a low molecular weight under 3000 Da, a minimum of 25% chondroitin sulphate; and 9% minerals, given as an indicative value.

The capsules used in this study were filled and identified with a code by a person who was not involved in the study. At the end of the study and after the statistical analysis, the code was revealed to the researchers, which guaranteed the double-blind study.

Participants of the Group 1—Placebo received supplementation containing hydrolyzed fish cartilage after the end of the study.

Both supplementations were well tolerated by all participants, without any reported side effects. In addition, all participants adhered to the treatment and reported that they took the supplements every day during the study period.

### 4.4. Instrumental Measurements

The instrumental measurements were performed after 20 min of acclimatization to an environment with controlled temperature between 20 °C to 22 °C and 45% to 55% relative humidity. The measurements were performed during the spring in the Southern Hemisphere (from September to December) at a clinical laboratory of the School of Pharmaceutical Sciences of Ribeirão Preto, University of São Paulo, Brazil (21°100′ S, 47°480′ W). The analyses were performed in the nasolabial region of the face before the beginning of the study (T0) and after 90 days (T90), according to the methods described below.

#### 4.4.1. Skin Microrelief

Skin microrelief was evaluated using the Visioscan^®^ VC 98 visiocamera and SELS 2000 software from Courage & Khazaka Electronic Gmbh (Cologne, Germany). This equipment provides qualitative and quantitative information about the surface of the skin under physiological conditions, using optical profilometry techniques based on an image digitization process obtained with a full video camera. The parameter evaluated was Sew (wrinkles)—calculated from the proportion of horizontal and vertical wrinkles and wrinkle depth [5,8,15].

#### 4.4.2. Dermis Echogenicity and Thickness

The 20 MHz ultrasound is a safe and noninvasive method for the assessment of changes in echogenicity and dermis thickness. Dermascan C^®^ (Cortex, Hadsund, Denmark) is one of the ultrasound analyzers available on the market for this purpose and is equipped with a transducer of strong focalization used for the capture of bi-dimensional and transverse images, represented by module-B in the software. The ultrasonic wave is partially reflected in the skin’s structures and generates echoes of different amplitudes. The intensity of the reflected echoes (echogenicity) is evaluated by a microprocessor and visualized as a two-dimensional color image. The color level of the echogenicity ranges from white to red > yellow> green > blue and black, with white being the most echogenic color and black the least echogenic one. For the calculation of echogenicity, the pixel number for low echogenicity is measured with software for image analysis and related to the total number of pixels [5,8,15].

#### 4.4.3. Skin Wrinkles and Pores Examined by High Definition Imaging Resolution

The Visioface^®^ digital photography imaging system (Courage & Khazaka, version 1.0.3.4) for the evaluation of facial skin consists of a cabin attached to a high-resolution digital camera (10 megapixels) and 200 white LED. This apparatus is connected to research software that permits the evaluation of visible spots (with the color image), pores, wrinkles, and color differences in the target area, which is selected manually [31,32,33].

The high-resolution images obtained were analyzed by a scoring method (Atlas Acervo NEATEC 2016) for wrinkles in the nasolabial, periorbital, and frontal regions of the skin and for pores in the malar region (Figure 11).

#### 4.4.4. Morphological and Structural Skin Characteristics Determined by Reflectance Confocal Microscopy

Morphological and structural characteristics of the epidermis and dermis were analyzed using a reflectance confocal microscope (Vivascope^®^ 1500; Lucid, New York, NY, USA). High resolution microscopic images (1000 × 1000 pixels) on the scale of 500 × 500 µm were produced in triplicate using the Vivastack^®^ system; these images represent multiple confocal images of successive depths taken within a given tissue region. Stacks from 1.52 μm to a depth of 25.91 μm and stacks from 3.05-μm to a depth of 108.2 μm were obtained from the skin surface. Stratum corneum (S.C.) and granular layer thickness; mean, minimum and maximum viable epidermal thickness; and dermal papilla depth were calculated using Vivastack^®^ images, representing the morphological features of each layer [2,4]. Mean epidermal thickness was calculated based on the mean of the maximum and minimum epidermal thickness. Total epidermal thickness was calculated based on the S.C. thickness plus maximum viable epidermal thickness. In addition, the RCM descriptors of photoaged skin were analyzed in terms of irregular honeycomb pattern, low interkeratinocyte reflectance, polycyclic papillary contours, effacement of the rete ridges, coarse and huddled collagen structures, and curled bright structures-elastosis [2,17].

Finally, the presence of thin reticulated collagen, coarse collagen, huddled collagen, and elastosis was also scored [17]. Score 1 represents the images with the presence of thin reticulated collagen, score 2 is the combination of thin reticulated collagen and coarse collagen, score 3 is the presence of coarse collagen, score 4 is the combination of coarse and huddled collagen with elastosis, and score 5 is the combination of huddled collagen with elastosis.

The RCM image analyses were performed before (T0) and after 90 days of treatment in the nasolabial region of the face of the seven participants in each study group. However, since some participants were lost to follow-up, only six participants per group were considered for analysis.

### 4.5. Perceived Efficacy

The participants were instructed to answer a questionnaire about the effects of the nutricosmetic product on skin improvement after treatment in terms of skin tonus, firmness, and hydration.

### 4.6. Statistical Analysis

Statistical analysis was performed using GraphPrism^®^ 8 and Origin^®^ 8 software. The Shapiro–Wilk test was used to determine normal distribution of the data. The paired Student t-test was used to compare results between two groups with parametric and dependent data, and the Wilcoxon test was used to compare the results of two groups with nonparametric and dependent data. One-way analysis of variance (ANOVA) followed by the Tukey test was used to compare results among three or more groups with parametric and independent data. The Kruskal–Wallis test followed by the Dunn test was used to compare the results of three or more groups with nonparametric and independent data. Finally, Fisher’s exact test was used to calculate the difference between categorical data. A *p* value < 0.05 was considered significant.

## Figures and Tables

**Figure 1 molecules-26-04880-f001:**
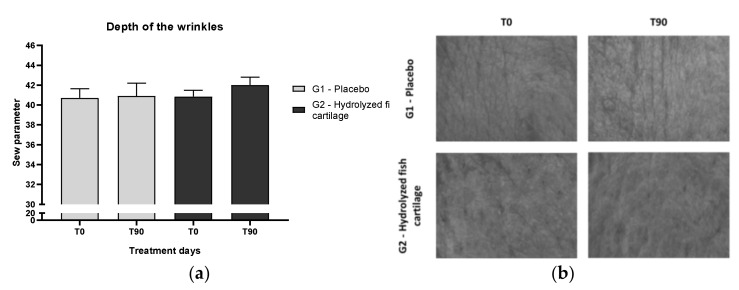
(**a**) Skin microrelief—Sew parameter (number and width of wrinkles) on the nasolabial region of the face, before (T0) and after 90 days (T90) of supplementation with hydrolyzed fish cartilage or placebo. (**b**) Representative skin microrelief images before (T0) and after 90 days (T90) of supplementation with hydrolyzed fish cartilage or placebo. Triplicates of images of each participant were analyzed using the Visioscan^®^ software.

**Figure 2 molecules-26-04880-f002:**
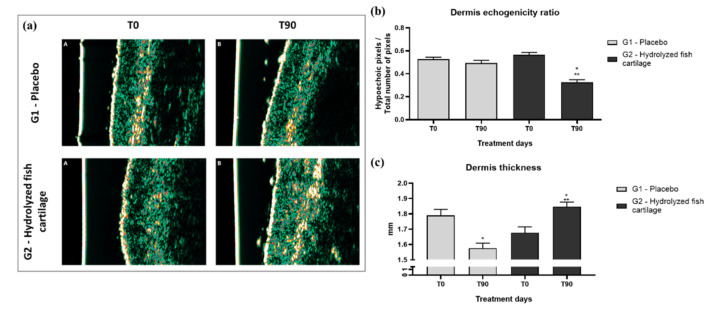
(**a**) Representative 20-MHz ultrasound images of dermis echogenicity before (T0) and after 90 days (T90) of supplementation with hydrolyzed fish cartilage or placebo. Echogenicity color scale: White > yellow > red > green > blue > black. (**b**) Dermis echogenicity ratio (hypoechogenic pixels/total number of pixels) before (T0) and after 90 days (T90) of supplementation with hydrolyzed fish cartilage or placebo. (**c**) Dermis thickness before (T0) and after 90 days (T90) of supplementation with hydrolyzed fish cartilage or placebo. * Significant differences when compared to baseline values—T0 (*p* < 0.05). ** Significant differences when compared to the placebo group.

**Figure 3 molecules-26-04880-f003:**
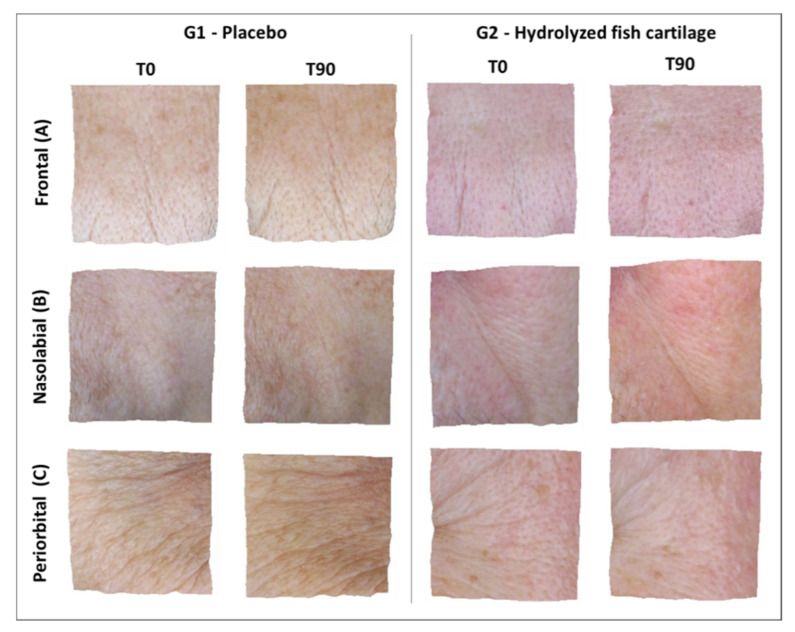
Representative 3D high-resolution images of wrinkles on the frontal (**A**), nasolabial (**B**), and periorbital (**C**) regions of the face before (T0) and after 90 days (T90) of supplementation with hydrolyzed fish cartilage or the placebo using the Visioface^®^ Quick software version 1.0.3.4.

**Figure 4 molecules-26-04880-f004:**
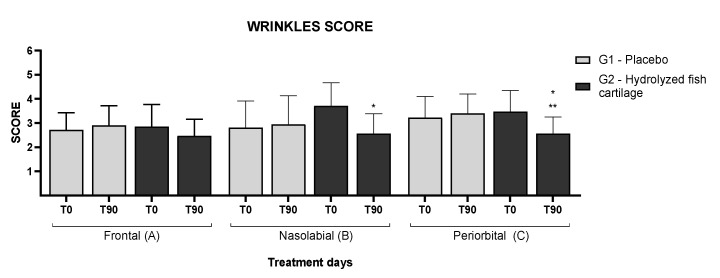
Wrinkle score in the frontal (**A**), nasolabial (**B**), and periorbital (**C**) regions of the face of the participants before (T0) and after 90 days (T90) of supplementation with hydrolyzed fish cartilage or the placebo. Triplicate images of each participant were analyzed using the Visioface^®^ Quick software version 1.0.3.4. * Significant differences when compared to baseline values—T0 (*p* < 0.05). ** Significant differences when compared to the placebo group.

**Figure 5 molecules-26-04880-f005:**
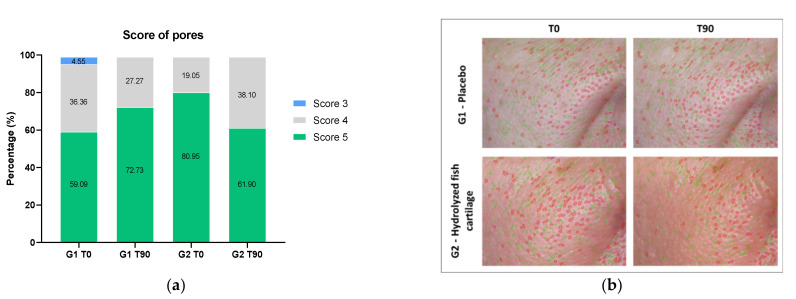
(**a**) Pore score (%) for the malar region of the face of the participants before (T0) and after 90 days (T90) of supplementation with the placebo or hydrolyzed fish cartilage. The results are reported as relative frequency of scores. (**b**) Representative 3D high-resolution images of pores in the malar region of the face before (T0) and after 90 days (T90) of supplementation with hydrolyzed fish cartilage or the placebo. Triplicate images of each participant were analyzed using the Visioface^®^ Quick software version 1.0.3.4.

**Figure 6 molecules-26-04880-f006:**
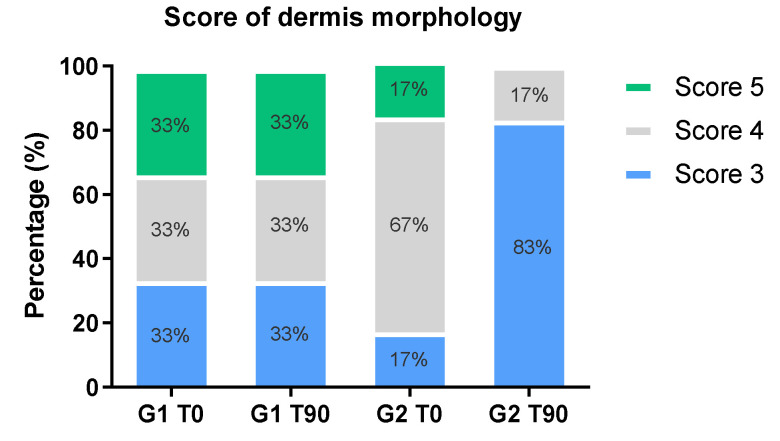
Score of dermis morphology (%) in the nasolabial region of the face of the participants before (baseline—T0) and after 90 days (T90) of supplementation with hydrolyzed fish cartilage or the placebo. The results are reported as relative frequency of scores. Triplicate images of each participant were analyzed using the Vivastack^®^ system.

**Figure 7 molecules-26-04880-f007:**
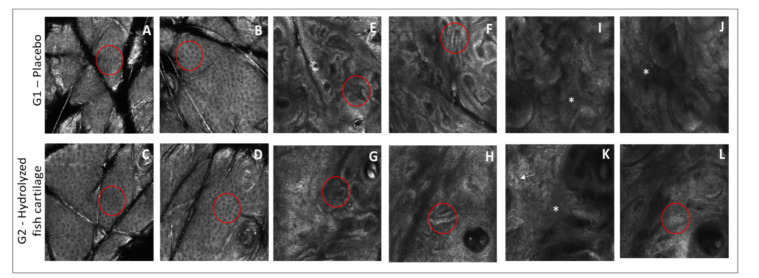
Representative reflectance confocal microscopy (RCM) images of the stratum granulosum of the skin. (**A**) Irregular honeycomb pattern and low interkeratinocyte reflectance (red circle) before supplementation with the placebo. (**B**) Regular honeycomb pattern and interkeratinocyte brightness (red circle) after supplementation with the placebo. (**C**) Irregular honeycomb pattern and low interkeratinocyte reflectance (red circle) before supplementation with hydrolyzed fish cartilage. (**D**) Regular honeycomb pattern and higher interkeratinocyte brightness (red circle) after supplementation with hydrolyzed fish cartilage. Representative RCM images of the dermal-epidermal junction. (**E**) Polycyclic papillary contours (red circle) before supplementation with the placebo; (**F**) polycyclic papillary contours (red circle) after supplementation with the placebo; (**G**) polycyclic papillary contours (red circle) before supplementation with hydrolyzed fish cartilage; (**H**) polycyclic papillary contours (red circle) after supplementation with hydrolyzed fish cartilage. Representative RCM images of the papillary dermis. (**I**) Huddled collagen structures (asterisks) before supplementation with placebo; (**J**) huddled collagen structures (asterisks) after supplementation with placebo; (**K**) huddled collagen structures (asterisks) with curled bright structures-elastosis (arrow) before supplementation with hydrolyzed fish cartilage; and (**L**) coarse collagen structures (red circle) after supplementation with hydrolyzed fish cartilage (Scale: 500 × 500 μm).

**Figure 8 molecules-26-04880-f008:**
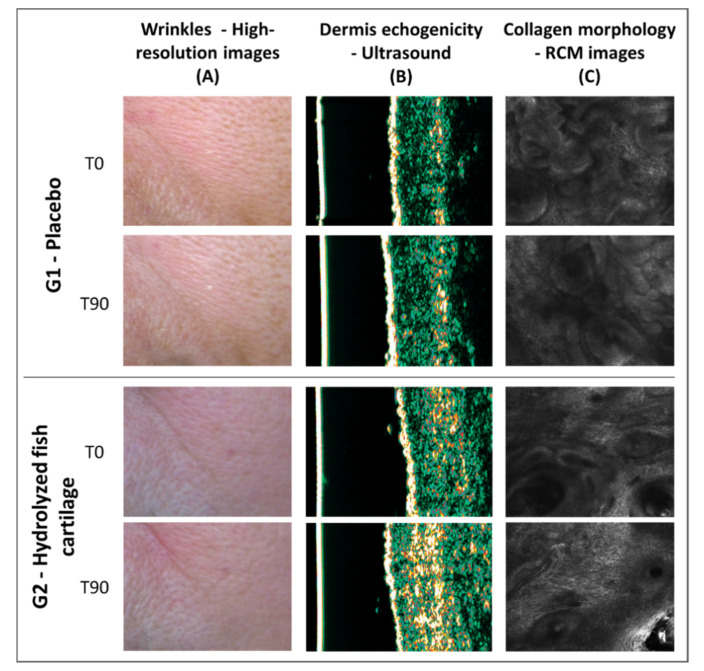
Representative images of skin showing wrinkle reduction (**A**), dermis echogenicity increase (**B**), and collagen morphology improvement (**C**) after a 90 day-period of supplementation with hydrolyzed fish cartilage (T90) compared to the baseline (T0) and to the placebo.

**Figure 9 molecules-26-04880-f009:**
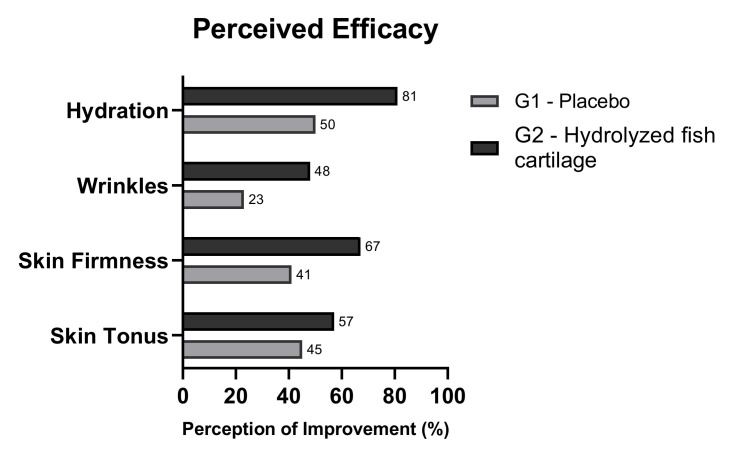
Perceived efficacy after both supplementations in terms of skin tonus, skin firmness, improvement of wrinkles, and hydration. The results are reported as relative frequency of categorical data.

**Figure 10 molecules-26-04880-f010:**
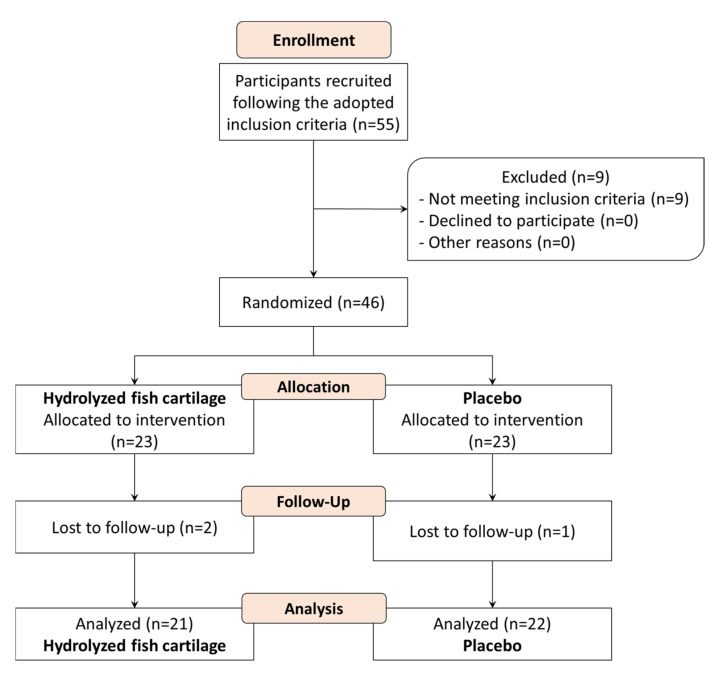
Flow diagram of participant selection for the study including dropout.

**Figure 11 molecules-26-04880-f011:**
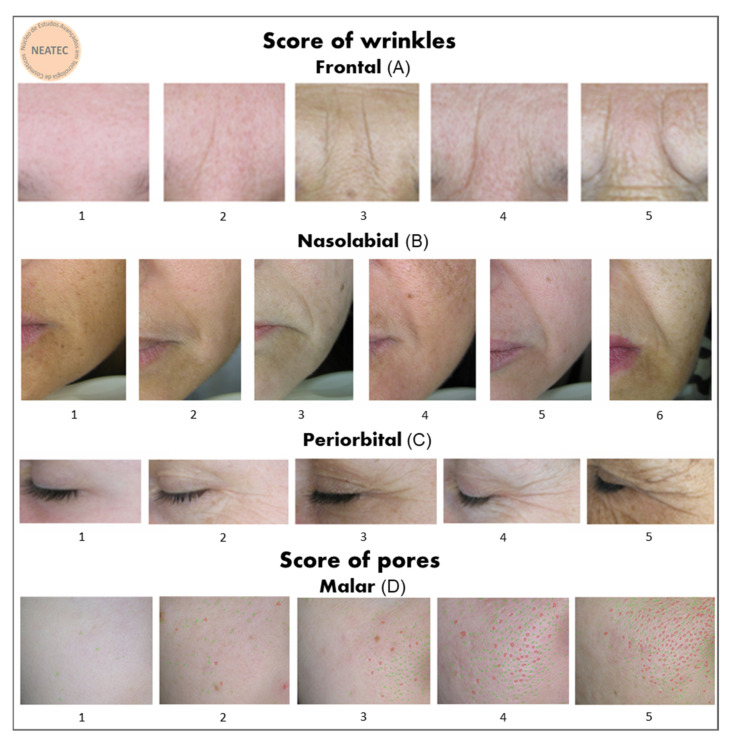
Wrinkle score in the frontal (**A**), nasolabial (**B**), and periorbital (**C**) regions of the skin and pore score in the malar (**D**) region of the skin obtained by NEATEC.

**Table 1 molecules-26-04880-t001:** Reflectance confocal microscopy (RCM) parameters of epidermis thickness.

Placebo Group (*n* = 6)				Hydrolyzed Fish Cartilage (*n* = 6)		
^1^ (µm)	T0	T90	*p* Value	T0	T90	*p* Value
Stratum Corneum Thickness	14.83 ± 1.86	16.33 ± 2.40	0.1362	15.00 ± 1.76	17.67 ± 1.33	0.0086 *
Granular Layer Thickness	17.00 ± 2.00	18.50 ± 2.35	0.3943	17.83 ± 1.83	20.50 ± 2.74	0.1055
Dermal Papilla Depth	17.50 ± 5.05	16.83 ± 3.37	1.0000	13.83 ± 5.00	13.47 ± 4.44	0.8287
Minimum Epidermis Thickness	21.17 ± 2.99	24.17 ± 2.71	0.0756	25.00 ± 5.97	28.50 ± 3.21	0.3072
Maximum Epidermis Thickness	38.67 ± 7.45	41.00 ± 4.94	0.3968	38.83 ± 7.88	41.97 ± 6.68	0.2777
Mean Epidermis Thickness	29.92 ± 5.08	32.58 ± 3.61	0.2296	31.92 ± 6.53	35.24 ± 4.75	0.2887
Total Epidermis Thickness	53.50 ± 8.58	57.33 ± 4.31	0.2647	53.83 ± 8.98	59.64 ± 7.33	0.0916

^1^ RCM parameters (µm) are reported as mean ± standard deviation. * Significant difference between T0 and T90.

**Table 2 molecules-26-04880-t002:** Reflectance confocal microscopy (RCM) descriptors of photoaged skin.

Placebo (*n* = 6)				Hydrolyzed Fish Cartilage (*n* = 6)		
RCM Descriptors	T0	T90	*p* Value	T0	T90	*p* Value
Irregular honeycomb pattern	67	33	0.0001 *	83	33	0.0001 *
Low interkeratinocyte reflectance	33	17	0.0138 *	33	0	0.0001 *
Polycyclic papillary contours	83	83	1.0000	67	67	1.0000
Effacement of rete ridges	17	17	1.0000	50	50	1.0000
Coarse collagen structures	67	67	1.0000	83	100	0.0001 *
Huddled collagen structures	33	33	1.0000	17	0	0.0001 *
Curled bright structures Elastosis	50	50	1.0000	83	67	0.0138 *

Results are reported as relative frequency of categorical data such as the presence of RCM descriptors (%). * Significant difference between times.

## Data Availability

The data presented in this study are available on request from the corresponding author and the permission of all parties involved in the study. The data are not publicly available due to privacy.

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
