# Peer review of "Oral Supplementation with Hydrolyzed Fish Cartilage Improves the Morphological and Structural Characteristics of the Skin: A Double-Blind, Placebo-Controlled Clinical Study"

_molecules, 2021, doi:10.3390/molecules26164880_

Round 1
Reviewer 1 Report
The article by Campos et al. entitled “Oral supplementation with hydrolyzed fish cartilage improves the morphological and structural characteristics of the skin: a double-blind, placebo-controlled clinical study” concerns studies on the influence of oral supplementation with fish cartilage on human skin appearance. The article concerns common and interesting topic in the field of cosmetology and provides valuable results. Authors describes double-blind placebo-controlled clinical study conducted in 46 healthy females (43 analyzed) which proved that 90-day oral supplementation of hydrolyzed fish cartilage caused a significant reduction of wrinkles and an increase of dermis echogenicity, as well as improved collagen morphology and reduced elastosis. However Authors should addressed some comments which are listed below:
- When describing results Authors should indicate if they compare G1 vs G2 or T0 vs T90 (Page 4 lines 6-13).
- When results are provided as percentages Authors should add some deviation indicators such as SEM or SD (Pages 4-6, Page 8, Table 2). Please also indicate if the given percentages are mean values?
- When describing results Authors should indicate the number of analyzed probes at least in the caption of the Figures 1a, 4, 5, 6.
- Why only 6 results are per tested group are given for the reflectance confocal microscopy? How those 6 participants were chosen from the groups of 21 or 22?
- In Abstract the abbreviation RCM should be defined eg. after reflectance confocal microscopy “(RCM)” should be added.
Author Response
The authors are deeply grateful for the comments and suggestions of the reviewers, which have greatly improved the article to be published in the journal. All changes were highlighted in the body of the manuscript using the "Track Changes" function in Microsoft Word.
The manuscript has been revised according to the suggestions and comments of the reviewers to improve the text.
Reviewer: 1
Comments and Suggestions for Authors
The article by Campos et al. entitled “Oral supplementation with hydrolyzed fish cartilage improves the morphological and structural characteristics of the skin: a double-blind, placebo-controlled clinical study” concerns studies on the influence of oral supplementation with fish cartilage on human skin appearance. The article concerns common and interesting topic in the field of cosmetology and provides valuable results. Authors describes double-blind placebo-controlled clinical study conducted in 46 healthy females (43 analyzed) which proved that 90-day oral supplementation of hydrolyzed fish cartilage caused a significant reduction of wrinkles and an increase of dermis echogenicity, as well as improved collagen morphology and reduced elastosis. However, Authors should addressed some comments which are listed below:
- When describing results Authors should indicate if they compare G1 vs G2 or T0 vs T90 (Page 4 lines 6-13).
The comparison was between the T0 and T90 of G2 group and only for the periorbital region, there was a significant difference compared to the placebo group. Indications for comparing the results were added to the manuscript, as suggested.
- When results are provided as percentages Authors should add some deviation indicators such as SEM or SD (Pages 4-6, Page 8, Table 2). Please also indicate if the given percentages are mean values?
The mean and standard deviation was added to the results provided as a percentage, as suggested. In addition, the percentages provided were reported as relative frequency of scores.
However, the results for Table 2 are reported as relative frequency of categorical data, such as the presence or absence of RCM descriptors. The caption in Table 2 has been changed to better explain the categorical data.
Also, the results for perceived efficacy (Figure 9) are reported as relative frequency of categorical data, such as the presence or absence of hydration, wrinkles, skin firmness and skin tonus and because of that we cannot calculate the mean and SD, as suggested. The caption in Figure 9 has been changed to better explain the categorical data.
- When describing results Authors should indicate the number of analyzed probes at least in the caption of the Figures 1a, 4, 5, 6.
In the results, we used only one probe in each one because the graphs are plotted from the analysis of different parameters from the images obtained. The equipment used and the number of analyzes were added to the caption.
The number of analyzed images and the software that was used to obtain these images were added to caption of Figures 1, 4, 5 and 6, as suggested.
4. Why only 6 results are per tested group are given for the reflectance confocal microscopy? How those 6 participants were chosen from the groups of 21 or 22?
The RCM analysis is very complex and generates multiple images that are obtained by the Vivastack® system. This Vivastack® system generates 45 images per region in depth ranging from 3 to 150 µm. The study protocol was to evaluate 3 Vivastack® system by a subject at each time. In total, there were 3240 images to be analyzed. For this reason, the RCM image analyses are more selective and with a smaller number of subjects to be able to correctly evaluate all the images obtained. In addition, the participants were randomly selected.
- In Abstract the abbreviation RCM should be defined eg. after reflectance confocal microscopy “(RCM)” should be added.
The definition of RCM (Reflectance Confocal Microscopy) was added to the abstract, as suggested.
Reviewer 2 Report
In this manuscript authors describe the clinical benefits of oral supplementation of collagen peptide derived from hydrolyzed fish cartilage for alleviating skin aging effects i.e reducing skin wrinkles, pores, maintenance of dermis echogenicity and thickness. They showed the effective outcomes of use of this peptide in terms of skin firmness, improved dermis density, and overall appearance - without any side effects. Manuscript provides detailed description of the outcomes, however, some minor suggestions are made for consideration by the author:
1. In this study, participants were advised to take capsule containing 500 mg of hydrolyzed fish cartilage per day. What is the basis of selection of dose of 500 mg specifically used for the study?.
2. Please mention the manufacturer information for the Maltodextrin ingested by the Placebo participants in the study.
3. In the discussion part (page 9, paragraph 7), authors state that "the benefits of oral supplementation with hydrolyzed fish cartilage may be associated with its bioavailability when compared to other collagen sources" This needs further explanation/elaboration.
This reviewer recommends acceptance of the manuscript with the above provisos.
Author Response
The authors are deeply grateful for the comments and suggestions of the reviewers, which have greatly improved the article to be published in the journal. All changes were highlighted in the body of the manuscript using the "Track Changes" function in Microsoft Word.
The manuscript has been revised according to the suggestions and comments of the reviewers to improve the text.
Reviewer: 2
Comments and Suggestions for Authors
In this manuscript authors describe the clinical benefits of oral supplementation of collagen peptide derived from hydrolyzed fish cartilage for alleviating skin aging effects i.e reducing skin wrinkles, pores, maintenance of dermis echogenicity and thickness. They showed the effective outcomes of use of this peptide in terms of skin firmness, improved dermis density, and overall appearance - without any side effects. Manuscript provides detailed description of the outcomes, however, some minor suggestions are made for consideration by the author:
1. In this study, participants were advised to take capsule containing 500 mg of hydrolyzed fish cartilage per day. What is the basis of selection of dose of 500 mg specifically used for the study?.
The dosage was chosen 500 mg/day regarding some clinical data in the literature on skin beauty, with similar type of products and low dosage:
- 750 mg/day – 8 weeks
- 500 mg/day – 3 months
- 600 mg/day - 180 days
In addition, we wanted to challenge benchmarks with high dosage of collagen type I (from 2,5 to 5g) showing skin benefits. In addition, we believe in the fact that our composition (collagen peptides + GAGs) is closer to the one of skin extracellular matrix (compared to collagen type I), so it could play a significant role in skin aging damages.
- Lassus, A.; Jeskanen,L.; Happonen, H. P.; Santalahti, J. Imedeen for the treatment of degenerated skin in females. J Int Med Res 1991, 19(2),147-52.
- Eskelinin, A.; Santalahti, J. Special natural cartilage polysaccharides for the treatment of sun-damaged skin in females. J Int Med Res 1992, 20(2),99-105.
- Ablon, G. A Double-blind, Placebo-controlled Study Evaluating the Efficacy of an Oral Supplement in Women with Self-perceived Thinning Hair. J Clin Aesthet Dermatol 2012, 5(11),28-34.
- Please mention the manufacturer information for the Maltodextrin ingested by the Placebo participants in the study.
The manufacturer for the Maltodextrin was Ingredion – Brazil.
- In the discussion part (page 9, paragraph 7), authors state that "the benefits of oral supplementation with hydrolyzed fish cartilage may be associated with its bioavailability when compared to other collagen sources" This needs further explanation/elaboration.
The paragraph was changed and added to manuscript:
The bioavailability of oral supplementation with collagen or hydrolysates depends on the type and source of this collagen1. In addition, bioavailability depends on the molecular weight of the collagen and marine sources generally have a low molecular weight, which could improve absorption2. Thus, the benefits of oral supplementation with hydrolyzed fish cartilage could be associated with its bioavailability when compared to other collagen sources2,3 .
- Ohara, H.; Matsuoto, H.; Ito, K.; Iwai, K.; Sato, K. Comparison of quantity and structures of hydroxyproline-containing peptides in human blood after oral ingestion of gelatin hydrolysates from different sources. J Agric Food Chem 2007, 55, 1532–1535.
- Vollmer, D.L.; West, V.A.; Lephart, E.D. Enhancing skin health: by oral administration of natural compounds and minerals with implications to the dermal microbiome. Int J Mol Sci 2018, 19, 3059–3094.
- Liu, D.; Nikoo, M.; Boran, G.; Zhou, P.; Regenstein, J.M. Collagen and gelatin. Annu Rev Food Sci Technol 2015, 6, 527–557.
Round 2
Reviewer 1 Report
The Authors addressed all comments and corrected their manuscript according to reviewer recommendations. The manuscript could be published in current form.
Author Response
The authors are deeply grateful for the comments and suggestions of the reviewer, which have greatly improved the article to be published in the journal. The manuscript has been revised according to the suggestions of the reviewers to improve the text.